

# Prevalence of depression and its impact on quality of life in frontline otorhinolaryngology nurses during the COVID-19 pandemic in China

Zi-Rong Tian[1,*], Xiaomeng Xie[2,3,*], Xiu-Ya Li[4], Yue Li[1], Qinge Zhang[5], Yan-Jie Zhao[2,3], Teris Cheung[6], Gabor S. Ungvari[7,8], Feng-Rong An[5] and Yu-Tao Xiang[2,3]

[1] Department of Nursing, Beijing Tongren Hospital, Capital Medical University, Beijing, China
[2] Department of Public Health and Medicinal Administration & Centre for Cognitive and Brain Sciences, University of Macau, Macau SAR, China
[3] Institute of Advanced Studies in Humanities and Social Sciences, University of Macau, Macau SAR, China
[4] Department of Otorhinolaryngology, Beijing Tongren Hospital, Capital Medical University, Beijing, China
[5] The National Clinical Research Center for Mental Disorders & Beijing Key Laboratory of Mental Disorders Beijing Anding Hospital & the Advanced Innovation Center for Human Brain Protection, Capital Medical University, School of Mental Health, Beijing, China
[6] School of Nursing, Hong Kong Polytechnic University, Hong Kong, SAR, China
[7] University of Notre Dame Australia, Fremantle, Australia
[8] Division of Psychiatry, School of Medicine, University of Western Australia, Perth, Australia
[*] These authors contributed equally to this work.

Corresponding authors
Feng-Rong An, afrylm@sina.com
Yu-Tao Xiang, YTXiang@um.edu.mo

## ABSTRACT

**Objective**. Exposure to the coronavirus disease 2019 (COVID-19) was associated with high risk of mental health problems among frontline nurses. This study examined the prevalence of depressive symptoms (depression hereafter) and its impact on quality of life (QOL) in otorhinolaryngology (ENT) nurses during the COVID-19 pandemic in China.

**Methods**. An online study was conducted between March 15 and March 20, 2020. Depression and QOL were assessed using standardized instruments.

**Results**. A total of 1,757 participants were recruited. The prevalence of depression was 33.75% (95% CI: 31.59%-35.97%). Results emerging from multiple logistic regression analysis showed that direct care of COVID-19 patients (OR: 1.441, 95% CI: 1.031–2.013, $P = 0.032$), and current smoking (OR: 2.880, 95% CI: 1.018–8.979, $P = 0.048$) were significantly associated with depression. After controlling for covariates, ENT nurses with depression had a lower overall QOL compared to those without depression ($F_{(1, 1757)} = 536.80$, $P < 0.001$).

**Conclusions**. Depression was common among ENT nurses during the COVID-19 pandemic in China. Considering the negative impact of depression on QOL and care quality, regular screening for depression should be conducted in ENT nurses and treatment should be provided.

## INTRODUCTION

The novel coronavirus disease (COVID-19) was first reported in Wuhan, China at the end of 2019. Since then the disease has been reported in more than 200 countries and territories, and COVID-19 has been declared a global public health emergency (*World Health Organization, 2020*). The reproduction number of COVID-19 ranges from 2.24 (95%CI [1.96–2.55]) to 3.58 (95%CI [2.89–4.39]) (*Zhao et al., 2020*). Similar to other respiratory viruses, this virus is spread mainly by respiratory droplets of infected cases when people speak, cough, or sneeze. In early phase of the COVID-19 outbreak, it was presumed that nosocomial transmission contributed to 41.3% of the infected patients in the general population and 29% of infected health care workers (*Wang et al., 2020*). By the nature of the clinical specialty, healthcare workers in otorhinolaryngology (ENT) units have a much higher likelihood to have direct contacts with COVID-19 patients compared with their counterparts in most of other clinical specialties. ENT nurses are exceptionally susceptible to aerosolized viral particles and high viral loads in the upper respiratory tract. This possibly explained why many health professionals working in ENT units were infected in the early stage of the COVID-19 outbreak (*Lu et al., 2020*). For example, in the UK an ENT consultant was the first frontline clinician who died on 30 March 2020 in combating COVID-19 (*Weaver, 2020*). Due to heavy clinical workload and high risk of infection, ENT nurses are more likely to suffer from psychological distress, which could increase the risk of more serious mental health problems, such as depression (*Venugopal, Mohan & Chennabasappa, 2020*; *Xu et al., 2020*).

Depression is associated with a range of negative health outcomes, such as increased risk of suicide, poor care quality and impaired occupational functions (*Gao et al., 2019*; *Knight, Air & Baune, 2018*; *Woo et al., 2016*). In order to reduce the risk of depression and develop appropriate preventive measures, it is important to understand its epidemiology. Quality of life (QOL) has been a widely used comprehensive health outcome in the past decades. To the best of our knowledge, there have been no studies examining the epidemiology of depression and its impact on QOL in ENT healthcare workers. Therefore, this study set out to examine the prevalence of depressive symptoms (depression hereafter) and their impact on QOL in frontline ENT nurses in China during the COVID-19 pandemic.

## MATERIALS & METHODS

### Setting and sample

This was a cross-sectional, anonymous online survey initiated by the Otolaryngology Branch, Chinese Nursing Association between March 15 and March 20, 2020 in China. Due to logistical reasons and the high risk of cross-infection, random sampling and face-to-face interviews were prohibited in almost all surveys involving frontline health professionals during the COVID-19 outbreak in China. Similar to other studies (*Lai et al., 2020*; *Zhang et al., 2020*), snowball sampling was used. The survey was conducted using the WeChat-based Questionnaire Star program. WeChat is a social communication application with over 1 billion users in China including all participants in this study. The Questionnaire Star program that has been widely used in many epidemiological surveys (*Li et al., 2016*;

*Liang & Fan, 2020*; *Xi & Liu, 2017*) was employed in this study. To be eligible, participants needed to be: (1) aged 18 years or above; (2) frontline nurses working in ENT units during the COVID-19 outbreak; (3) able to understand the assessment and provide written informed consent. The research protocol was approved by the Institutional Review Board of Beijing Anding Hospital (2020(10)). All the study procedures were carried out in accordance with relevant guidelines. All participants provided informed consent to participate in the study.

## Instruments

Basic socio-demographic and clinical variables, such as gender, age, marital status, education level, years of work experience, living circumstances, rank (junior or senior), hospital setting (primary/secondary vs tertiary hospitals), shift duty requirement, type of the unit (inpatient or outpatient), smoking status, and personal experience with the Severe Acute Respiratory Syndrome (SARS) outbreak on 2003 were collected. Three additional standardized questions with dichotomous response (yes/no) were also asked: (1) whether the participant was directly engaged in clinical services for COVID-19 patients; (2) whether they had friends, colleagues, or family members infected with COVID-19; and (3) whether the number of COVID-19 confirmed cases in the province where they lived exceeded 500.

The self-report Chinese version of the Patient Health Questionnaire-9 (PHQ-9) was used to measure the severity of depression in the past week. The PHQ-9 was validated in Chinese populations with a sensitivity of 0.89 and a specificity of 0.77 (*Chen, Sheng & Qu, 2015*). Each item was scored from 0 to 3, with the total score of $\geq 5$ indicating "depression" (*Wittkampf et al., 2007*). The total score of "5-9", "10-14", "15-19", and "20-27" indicated "mild depression", "moderate depression", "moderate-to-severe depression", and "severe depression", respectively (*Wittkampf et al., 2007*).

Following the example of previous studies (*An et al., 2020*; *Ma et al., 2020*; *Wang et al., 2006*) QOL was estimated with the first two items on overall QOL of the validated World Health Organization Quality of Life Instrument-Brief Version (WHOQOL-BREF) (*Skevington et al., 2004*). Higher total scores indicated better QOL. The Chinese version of the WHOQOL-BREF has been validated in Chinese populations (*Xia et al., 2012*).

## Data analysis

Data were analyzed with the IBM Statistical Package for Social Science (SPSS), software version 24.0. Normality of the data was assessed using the Kolmogorov–Smirnov test. Comparison between the 'depression' and 'no depression' groups in terms of demographic and clinical characteristics were conducted by chi-square test, two independent samples $t$-test and Mann–Whitney U test, as appropriate. QOL was compared between the two groups using analysis of covariance (ANCOVA) after controlling the potentially confounding effects of variables with significant group difference in univariate analyses. The independent demographic and clinical correlates of depression were examined using multiple logistic regression analysis with the "Enter" method with depression as the dependent variable. All variables with a $P$-value of less than 0.1 in univariate analyses were entered as independent variables. A $P$-value of less than 0.05 was considered statistically significant (two-tailed).

## RESULTS

A total of 1,757 frontline ENT nurses (females $n = 1,746$, 99.4% of the sample) participated in the study. The overall prevalence of depression was 33.75% (95% CI [31.59%–35.97%]). Among the healthcare workers with probable depression ($N = 593$), 421 (24.0%) reported mild, 116 (6.6%) moderate, 42 (2.4%) moderate-to-severe, and 14 (0.8%) severe depression. The mean total score of the PHQ-9 scale was 3.73 ($\pm4.43$) in the whole sample.

Table 1 shows the demographic and clinical characteristics of the whole sample separated by depression. Univariate analyses revealed that direct care with confirmed COVID-19 patients ($P = 0.025$), current smoking ($P = 0.033$), and years of work experience ($P = 0.020$) were significantly associated with depression. After controlling for covariates including looking after infected patients, smoking, work experience, depressed nurses were more likely to have overall lower QOL than those without depression ($F_{(1,1757)} = 527.94$, $P<0.001$). Five variables with a $P$-value of <0.1 were entered in multiple logistic regression analysis as independent variables including working in tertiary hospitals, looking after infected patients, current smoking, age, and work experience. Direct care of COVID-19 patients (OR $= 1.441$, $P = 0.032$) and smoking (OR $= 2.880$, $P = 0.048$) were independently associated with higher risk of depression (Table 2).

## DISCUSSION

To the best of our knowledge, this was the first study that examined the prevalence, demographic and clinical factors associated with depression in ENT nurses during the COVID-19 pandemic. Other studies have examined the epidemiology of depression in health professionals in China. In the early stage of the COVID-19 outbreak at the end of January 2020, 50.4% of frontline medical professionals working in Wuhan and the surrounding areas of Hubei province reported depression measured using the PHQ-9 with a cut-off value of 5 (*Lai et al., 2020*). With the same cut off value in the PHQ-9, the prevalence of depression in healthcare workers in Wuhan was 36.9% between January 29 and February 4, 2020 (*Kang et al., 2020*). In contrast, the prevalence of depression in frontline healthcare workers was 12.2% assessed with the PHQ-2 with a lower cut-off value of 3 from February 19 to March 6, 2020 (*Zhang et al., 2020*). Findings of the current study (33.75%; 95% CI: 31.59%-35.97%) were similar to those of some (*Kang et al., 2020*), but not all studies (*Lai et al., 2020*; *Zhang et al., 2020*). Due to the use of different measurement tools on depression, direct comparison between these studies should be interpreted with caution.

In ENT units, asymptomatic and pre-symptomatic patients with COVID-19 may seek help for anosmia, which is a common symptom of COVID-19 (*Hopkins, Surda & Kumar, 2020*). Examinations of the nasal passages and upper airway, intubation and administration of respiratory treatment can induce cough, nausea, sneezing or vomiting (*Lu et al., 2020*). The nasal pillow masks for patients with obstructive sleep-apnea may allow airborne virus transmission due to loose sealing (*Tran et al., 2012*). In the 2003 SARS outbreak, clusters of nosocomial infections were observed among healthcare workers during airway manipulation (*Morbidity and Mortality Weekly Report (2003)*). All these factors could

**Table 1 Demographic characteristics of the study cohort of ENT nurses.**

| Variables | Total (N = 1,757) | | No depression (N = 1,164) | | Depression (N = 593) | | X² | df | P |
|---|---|---|---|---|---|---|---|---|---|
| | N | % | N | % | N | % | | | |
| Married | 1310 | 74.6 | 875 | 75.2 | 435 | 73.4 | 0.683 | 1 | 0.409 |
| College education and above | 1707 | 97.2 | 1131 | 97.2 | 576 | 97.1 | 0.001 | 1 | 0.970 |
| Living with family | 1483 | 84.4 | 988 | 84.9 | 495 | 83.5 | 0.590 | 1 | 0.442 |
| Junior rank | 1017 | 57.9 | 683 | 58.7 | 334 | 56.3 | 0.892 | 1 | 0.345 |
| Experience with SARS | 204 | 11.6 | 136 | 11.7 | 68 | 11.5 | 0.018 | 1 | 0.893 |
| Working in tertiary hospitals | 1528 | 87.0 | 1001 | 86.0 | 527 | 88.9 | 2.862 | 1 | 0.091 |
| Working in inpatient department | 1535 | 87.4 | 1024 | 88.0 | 511 | 86.2 | 1.154 | 1 | 0.283 |
| Shift duty | 1195 | 68.0 | 789 | 67.8 | 406 | 68.5 | 0.084 | 1 | 0.772 |
| Local COVID-19 cases ≥ 500 | 235 | 13.4 | 161 | 13.8 | 74 | 12.5 | 0.620 | 1 | 0.431 |
| Having infected family/friends/colleagues | 86 | 4.9 | 50 | 4.3 | 36 | 6.1 | 2.660 | 1 | 0.103 |
| Looking after infected patients | 158 | 9.0 | 92 | 7.9 | 66 | 11.1 | 4.996 | 1 | **0.025** |
| Current smoker | 13 | 0.7 | 5 | 0.4 | 8 | 1.3 | 4.523 | 1 | **0.033** |
| | Mean | SD | Mean | SD | Mean | SD | T/Z | df | P |
| Age (years) | 34.09 | 8.03 | 33.86 | 8.23 | 34.56 | 7.62 | −1.729 | 1755 | 0.084 |
| Work experience (years) | 12.72 | 8.82 | 12.51 | 9.01 | 13.12 | 8.42 | −2.324 | —[a] | **0.020** |
| Total QOL score | 6.64 | 1.57 | 7.18 | 1.36 | 5.58 | 1.40 | 23.064 | 1755 | **<0.001** |

Notes.

Note: due to the very small sample size of male nurses in this sample (N = 11), gender was not included in analyses.

[a]Mann–Whitney U test

Bolded values: $P < 0.05$

M, mean; SD, standard deviation; COVID-19, Corona Virus Disease 2019; SARS, Severe Acute Respiratory Syndrome; QOL, Quality of Life.

**Table 2 Independent correlates of depression by multiple logistic regression analysis.**

| Variables | Multiple logistic regression analysis | | |
|---|---|---|---|
| | OR | 95% CI | P value |
| Working in tertiary hospitals | 1.295 | 0.953-1.761 | 0.098 |
| Looking after infected patients | 1.441 | 1.031-2.013 | **0.032** |
| Current smoker | 2.880 | 1.018-8.979 | **0.048** |
| Age (years) | 1.028 | 0.984-1.074 | 0.216 |
| Work experience (years) | 0.984 | 0.945-1.024 | 0.423 |

Notes.

Note: No collinearity between independent variables was found. Bolded values: $P < 0.05$

CI, confidential interval; OR, odds ratio; QOL, Quality of Life.

increase the likelihood of COVID-19 infection for ENT nurses, leading to common mental health problems, such as depression.

Similar to previous findings (*Lai et al., 2020*; *Pan et al., 2020*), frontline ENT nurses who provided direct care for COVID-19 patients were more likely to have depression. During the COVID-19 outbreak, ENT nurses had to do shift duty and worked longer hours than usual, which can lead to high level of stress. All health professionals were confined to at least two weeks quarantine after they finished care to COVID-19 patients, which can increase

their anxiety and induce guilt feelings due to the social stigma affecting their families, as it happened during the SARS epidemic (*Holmes et al., 2020*; *Li et al., 2020*; *Nickell et al., 2004*; *Yip et al., 2010*). All these factors could substantially increase the risk of depression. Smoking is associated with higher risk of medical conditions and psychiatric disturbances including depression (*Chang, Lau & Moolgavkar, 2020*; *Fluharty et al., 2017*; *Mathew et al., 2017*). This study also found that depressed ENT healthcare workers were more likely to smoke (*Nilan et al., 2019*; *Schneider et al., 2019*).

According to the distress/protection model of QOL (*Voruganti et al., 1998*), QOL is the result of the interaction between protective (e.g., high self-esteem and good social support) and distressing factors (e.g., physical and psychological stress). Consistent with previous findings (*Benedek, Fullerton & Ursano, 2007*; *Mammen & Faulkner, 2013*; *Roche et al., 2020*) depressed ENT nurses had a poorer QOL compared to the those without depression in this survey. This could be explained by the negative health outcomes of depression, such as impaired psychosocial functioning and somatic symptoms of fatigue, loss of appetite or weight, and insomnia (*Anosike, Isah & Igboeli, 2020*; *Malhi & Mann, 2018*; *Parisi et al., 2014*; *Rakofsky et al., 2013*).

The strengths of this study include the large sample size and the use of standardized instruments. However, several limitations should be addressed. First, because of the cross-sectional design, the causality between the demographic and clinical variables and depression could not be established. Second, data were collected by online self-report, therefore the the validity of certain variables (e.g., history of psychiatric disorders) could not be tested and recall bias could exist, which is a common limitation in all online surveys. Third, due to logistical reasons, relevant factors related to depression in ENT nurses, such as lifestyle, family support, work load including the number of daily outpatient visits and inpatients in participating hospitals, were not obtained. Forth, due to the lack of rating scales on COVID-19-related experiences in China, participants were asked only using three standardized questions with dichotomous response, similar to previous studies (*Forte et al., 2020*; *Zhong et al., 2020*). Fifth, the snowball sampling method was used, thus the number of ENT nurses who did not complete the assessment or refused to participate in the study could not be recorded. Finally, this is an anonymous survey, therefore, participants' hospital and personal identifying information was not collected. Sixth, the exclusion of participants with pre-existing mental health problems could have biased the results to an uncertain extent.

## CONCLUSIONS

Depression was common among ENT nurses during the COVID-19 pandemic in China. Considering the negative impact of depression on their QOL and the quality of care ENT nurses provide, regular screening for depression should be conducted for this particularly vulnerable cohort of health workers coupled with easily available treatment.

### Funding

The study was supported by the National Science and Technology Major Project for investigational new drug (2018ZX09201-014), the Beijing Municipal Science & Technology Commission (No. Z181100001518005), and the University of Macau (MYRG2019-00066-FHS). The funders had no role in study design, data collection and analysis, decision to publish, or preparation of the manuscript.

### Grant Disclosures

The following grant information was disclosed by the authors:
National Science and Technology Major Project: 2018ZX09201-014.
Beijing Municipal Science & Technology Commission: Z181100001518005.
University of Macau: MYRG2019-00066-FHS.

### Competing Interests

The authors declare there are no competing interests.

### Author Contributions

- Zi-Rong Tian, Xiaomeng Xie and Xiu-Ya Li performed the experiments, prepared figures and/or tables, authored or reviewed drafts of the paper, and approved the final draft.
- Yue Li performed the experiments, authored or reviewed drafts of the paper, and approved the final draft.
- Qinge Zhang, Yan-Jie Zhao, Teris Cheung and Gabor S. Ungvari analyzed the data, authored or reviewed drafts of the paper, and approved the final draft.
- Feng-Rong An and Yu-Tao Xiang conceived and designed the experiments, authored or reviewed drafts of the paper, and approved the final draft.

### Human Ethics

The following information was supplied relating to ethical approvals (i.e., approving body and any reference numbers):

The Institutional Review Board of Beijing Anding Hospital approved this research (2020(10)).

### Data Availability

Raw measurements are available in the Supplemental Files.

### Supplemental Information

Supplemental information for this article can be found online at http://dx.doi.org/10.7717/peerj.11037#supplemental-information.

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
