# Peer review of "Prevalence of depression and its impact on quality of life in frontline otorhinolaryngology nurses during the COVID-19 pandemic in China"

_PeerJ, doi:10.7717/peerj.11037_

## Round 0.1 · original submission · Major Revisions

Thank you for submitting your interesting manuscript. The reviewers of your manuscript recommended major revisions. Please kindly address these issues accordingly.

Reviewer 1 ·

Basic reporting

None

Experimental design

None

Validity of the findings

None

Additional comments

This is an online survey, which aimed to examine the prevalence of depressive symptoms and its association with quality of life among frontline otorhinolaryngology (ENT) nurses in China during the COVID-19 pandemic. Though this is a well-organized manuscript, I have some comments for the authors for their consideration.
1. Authors mentioned that this is a national survey. However, subjects are from the online platform. How to define the “national survey” in this article?
2. Three additional questions were used, but not be validated. The Cronbach alpha value should be considered.
3. why do authors only use the first two items of WHOQOL- BREF? This is not a common way.
4. Authors mentioned that a total of 1,757 frontline ENT nurses participated in this study. However, I would like to know the response rate or completing rate. How many people refused to answer this survey? or How many questionnaires were eliminated due to the certain reasons, such as logistic errors, incomplete questionnaires?
5. Authors should describe the specific covariates. Please add them.
6. How do authors consider the collinearity? Such as a potential correlation between junior clinicians and working experience
7. Based on the criterion of P< 0.1, why not include the age (p=0.084) in the multiple logistic regression analysis.
8. Authors did not mention gender in this manuscript. I guess that all samples are female. But authors should describe this part.

Reviewer 2 ·

Basic reporting

no comment

Experimental design

The representativeness of the sample for otorhinolaryngology nurses during the COVID-19 pandemic in China is unclear, because there is no sampling frame. Selection bias exist.

Validity of the findings

no comment

Additional comments

This survey used snowball sampling and WeChat-based Questionnaire Star program and investigated 1,757 frontline otorhinolaryngology nurses on depression symptoms and quality of life. Prevalence of depression symptoms in this population was estimated. However, I have several concerns as follows:
1. The representativeness of the sample for otorhinolaryngology nurses during the COVID-19 pandemic in China is unclear, because there is no sampling frame. How many provinces and hospitals involved? Selection bias exist.
2. Less information on quality control was provided. How did the investigators verify those who WeChat-based Questionnaire were otorhinolaryngology nurses? In adidition, because people with mental health problem tend to take part in this problem, is there any bias on this sample?
3. The online survey initiated by the Chinese Nursing Association Otolaryngology Branch between March 15 and March 20, in China. Actually from January, China has introduce a series of policies on the prevention and control of COVID-19 in each level of hospitals. Is there any evidence on the daily outpatient visit or the number of hospitalized patients in the hospitals and otolaryngology department? Because this may influence the clinical workload of the ENT units.
4. Besides primary and tertiary hospital settings, is there second-class hospitals included?
5. The authors used PHQ-9 to measure the severity of depression. The duration measured was within 2 weeks, which should be clarified.
6. The authors said "After controlling for covariates, nurses with depression were more likely to have overall lower QOL than those without". The covariates which controlled should be displayed. Besides, is this result showed in Table 1?
7. Table 2 showed the results of multiple logistic regression analysis which examine the Independent correlates of depression. But I did not find the QOL in this model.
8. "All health professionals must have at least two weeks quarantine after they finished providing care to COVID-19 patients, which may put them in anxiety state and guilty feelings due to social stigma on their families." Is there any reference for this?

---

## Round 0.2 · Minor Revisions

Please address the two minor issues accordingly.

Reviewer 1 ·

Basic reporting

none

Experimental design

none

Validity of the findings

none

Additional comments

none

Reviewer 2 ·

Basic reporting

no comment

Experimental design

no comment

Validity of the findings

no comment

Additional comments

Thank the authors for considering my suggestions. I have several concerns before the publication:
1. The authors mentioned "the exclusion of participants with pre-existing mental health problems", how did they excluded participants with pre-existing mental health problems? On the contrary, I guess people with mental health problems were more likely to take part in this kind of survey, because they could be more concerned about their mental health.
2. The authors said "the data were collected online using smallball sampling with the Questionnaire Star program which has been widely used in many epidemiological studies. Therefore, certain variables, such as ... and participating hospitals, could not be recorded or calculated". Actually it is a limitation in the survey Questionnaire designing. They could have added items for information on participating hospitals and even the identity of participants.

---

## Round 0.3 · accepted · Accept

The authors have appropriately addressed the remaining issues.